# Bring More Data!—A Good Advice? Removing Separation in Logistic Regression by Increasing Sample Size

**DOI:** 10.3390/ijerph16234658

**Published:** 2019-11-22

**Authors:** Hana Šinkovec, Angelika Geroldinger, Georg Heinze

**Affiliations:** Institute of Clinical Biometrics, Center for Medical Statistics, Informatics and Intelligent Systems (CEMSIIS), Spitalgasse 23, 1090 Vienna, Austria; hana.sinkovec@meduniwien.ac.at (H.Š.); angelika.geroldinger@meduniwien.ac.at (A.G.)

**Keywords:** maximum likelihood estimation, logistic regression, Firth’s correction, separation, penalized likelihood, bias

## Abstract

The parameters of logistic regression models are usually obtained by the method of maximum likelihood (ML). However, in analyses of small data sets or data sets with unbalanced outcomes or exposures, ML parameter estimates may not exist. This situation has been termed ‘separation’ as the two outcome groups are separated by the values of a covariate or a linear combination of covariates. To overcome the problem of non-existing ML parameter estimates, applying Firth’s correction (FC) was proposed. In practice, however, a principal investigator might be advised to ‘bring more data’ in order to solve a separation issue. We illustrate the problem by means of examples from colorectal cancer screening and ornithology. It is unclear if such an increasing sample size (ISS) strategy that keeps sampling new observations until separation is removed improves estimation compared to applying FC to the original data set. We performed an extensive simulation study where the main focus was to estimate the cost-adjusted relative efficiency of ML combined with ISS compared to FC. FC yielded reasonably small root mean squared errors and proved to be the more efficient estimator. Given our findings, we propose not to adapt the sample size when separation is encountered but to use FC as the default method of analysis whenever the number of observations or outcome events is critically low.

## 1. Introduction

In medical research, logistic regression is a popular method used to study the relationship between a binary outcome and a set of covariates. Regression coefficients can be interpreted as log odds ratios and are usually estimated by the method of maximum likelihood (ML). Moreover, individualized prognosis can be obtained by estimating the probability of an outcome given the covariates, making logistic regression widely used in the era of personalized medicine.

Despite analytically attractive properties under some regulatory conditions as the sample size increases [1,2], in analyses of small or sparse data sets, the properties of ML estimator become questionable: ML coefficient estimates are biased away from zero and very unstable [3] or may even not exist [4]. The situation in which the ML estimate of at least one regression coefficient does not exist, i.e., diverges to plus or minus infinity, has been termed ‘separation’ as the two outcome groups are separated by the values of a covariate or a linear combination of covariates. Therefore, perfect predictions for some (quasi-complete separation) or for all observations (complete separation) of the data set the model is fitted on are obtained. The simplest case of separation arises when an odds ratio is estimated from a 2×2 contingency table with one zero cell count. In addition to a small sample size and sparsity various other factors, such as a small relative frequency of one of two levels of the outcome variable (unbalanced outcome), rare exposures and strong associations with the outcome can give rise to separation [5,6]. A large number of strongly correlated covariates is another such factor that has induced a possibly unjustified promotion of the 10 events per variable rule in biomedical literature [7,8].

In order to solve the problem of non-existing ML coefficient estimates, applying Firth’s correction (FC) [9], originally intended for bias reduction in generalized linear models, was proposed [5]. FC was shown to be robust to the problem of separation, making it a default choice in situations where separation is likely to occur. Although separation could also arise as a feature of the data-generating process, we are interested in situations where separation occurs as a sampling artefact, which is a consequence of random sampling variation and can be removed by increasing sample size. Therefore, in such a case, a principal investigator is often advised to ‘bring more data’ in order to solve a separation issue. It is unclear if such a simple increasing sample size (ISS) strategy that keeps sampling new observations until separation is removed improves estimation compared to FC applied to the original data set.

The objective of this paper is therefore to investigate the performance of ML after separation has been removed by ISS compared to FC applied to the initial data. The paper is organized as follows. In the next section, we briefly summarize the logistic regression model together with the ML and the FC estimators. In addition, we provide a simple example illustrating the separation problem and describe the setup for a simulation study. Subsequently, we describe our findings evaluating the empirical performance of ML combined with ISS and FC. As illustrative examples, we consider preliminary and final analyses of two studies from colorectal cancer screening and from ornithology. Finally, we summarize our most important findings.

## 2. Materials and Methods

### 2.1. General

Assume that we have N independent observations $\left(x_{i},\;y_{i}\right),\;i=1,\ldots ,\;N$, where $y_{i}\in \left\{0,1\right\}$ denotes a binary outcome with level 1 occurring with probability $\pi_{i}$ and $x_{i}=\left(x_{i1}\ldots x_{iK}\right)$, K<N, denotes a vector of covariate values for the i-th subject. With logistic regression, designed to provide individualized prognosis ${\hat{\pi }}_{i}$ given $x_{i}$ by ensuring that ${\hat{\pi }}_{i}\in \left[0,1\right]$ [10], the logit (i.e., the log-odds) of $\pi_{i}$ is modeled as a linear combination of the covariate values:(1)$$\log\left(\frac{\pi_{i}}{1-\pi_{i}}\right)=\beta_{0}+\beta_{1}x_{i1}+\ldots +\beta_{K}x_{iK},$$
where $\beta_{0}$ is an intercept and $\beta_{k}$, k=1,…K, are regression coefficients for the covariates $X_{k}$. The regression coefficients for covariates $X_{k}$ can be interpreted as log odds ratios, corresponding to one-unit differences in $X_{k}$. They are usually obtained by ML estimation, maximizing the log-likelihood function:(2)$$l\left(\beta \right)=\overset{N}{\underset{i=1}{\sum }}y_{i}\log\pi_{i}+\left(1-y_{i}\right)\log\left(1-\pi_{i}\right).$$

In the simplest case with one binary covariate, X observations can be classified according to x and y in a 2×2 contingency table with observed frequencies $f_{00},\;\;f_{01,},\;f_{10},\;f_{11}:$



x

01y0
$f_{00}$

$f_{01}$
1
$f_{10}$

$f_{11}$


The ML estimate of β is then given by:(3)$$\hat{\beta }=\log\left(\frac{f_{00}f_{11}}{f_{01}f_{10}}\right),$$
and predicted probabilities are equal to the proportions of events in the two groups.

Now suppose that at least one of the observed frequencies of the 2×2 table, e.g., $f_{11}=0$, is zero. Accordingly, the ML estimate of the predicted probability for x=1 is zero, while the ML regression coefficient $\hat{\beta }$ is not defined. According to Equation (3), whenever one of the cell counts of a 2×2 table is zero, $\hat{\beta }$ is not defined. To obtain a finite regression coefficient, we have to assume ${\hat{\pi }}_{i}\in \left(0,1\right)$. This is in accordance with our general assumption on separation being a sampling artefact, which can always be removed by increasing sample size. If so, a finite number of covariates can never be sufficient to perfectly predict an outcome in the underlying population.

In FC, the likelihood function is penalized by the Jeffreys invariant prior [9]. For 2×2 tables, FC is equivalent to ML after adding a 0.5 to each of the observed frequencies [11]. Applying FC does not only remove the first-order bias from ML regression coefficient estimates in generalized linear models [9], which may be severe in small or sparse data sets. Moreover, FC is robust to the problem of separation [5], as predicted probabilities are pulled to 0.5 such that ${\hat{\pi }}_{i}\in \left(0,1\right)$. Note that other methods imposing shrinkage on the regression coefficients, which are often based on weakly or strongly informative priors, can as well overcome the separation issue [11,12,13].

### 2.2. Simulation Study Setup

The simulation study setup is described as recommended by Morris [14].

Aims: We intended to systematically investigate the performance of ML after removing separation by ISS (ML+ISS) in comparison to FC applied to the original data, focusing on scenarios where separation is likely to occur. We also investigated the performance of combining FC with ISS (FC+ISS).

Data-generating mechanisms: To capture a context plausible in medical research, we considered a data-generating scheme as described in Binder et al. [15]. First, we generated data sets of size N with a binary outcome variable Y and K covariates $X_{k}$ of mixed types that were obtained by applying certain transformations to variables $Z_{1},\;\ldots ,Z_{10}$ sampled from a standard multivariate normal distribution with correlation matrix Σ, see Table 1. In this way, we obtained four binary covariates $X_{1},\;\ldots ,X_{4}$, two ordinal covariates, $X_{5}$ and $X_{6}$, with three levels, and four continuous covariates $X_{7},\;\ldots ,X_{10}$ (Table 1). In order to avoid extreme values, we truncated continuous covariates at the third quartile plus five times the interquartile distance of the corresponding distribution.

We considered a full factorial design, varying the number of covariates $K\in \left\{2,\;5,\;10\right\}$, the sample size $N\in \left\{80,\;200,\;500\right\},$ the expected value of Y,
$E\left(Y\right)\in \left\{0.1,\;0.25\right\},$ and the value of $\beta_{1}\in \left\{0,\;0.35,\;1.39,\;2.77\right\}$. This resulted in 72 possible combinations of simulation parameters. We simulated 1000 data sets with each of those combinations. We held the true regression coefficients of covariates $X_{2},\;\ldots ,X_{10}$ constant across simulation scenarios, setting them to log2=0.69 for binary covariates, and to log√2=0.35 for ordinal covariates. For continuous covariates, the effects were chosen such that the log odds ratio between the first and the fifth sextile of the corresponding distribution was 0.69. An intercept $\beta_{0}$ was determined for each simulation scenario such that the desired proportion of events was approximately obtained. Finally, we sampled the binary outcome $y_{i}$ for subject i, i=1,…N, from a Bernoulli distribution as $y_{i}\sim \mathrm{Bern}\left(\pi_{i}\right)$ after calculating $\pi_{i}$ by $\pi_{i}=1/\left(1+\exp\left(-\beta_{0}-\beta_{1}x_{i1}-\ldots -\beta_{K}x_{iK}\right)\right)$. Whenever separation was encountered in the original data set of size N, we added to these data new observations sampled from the same distribution to generate an increased data set of sample size $N_{new}$ in which separation was removed.

Methods: We analyzed each simulated data set by fitting a logistic regression model (1) and estimating the regression coefficients by:ML after removing separation by ISS (ML+ISS);FC applied to the original data; andFC after removing separation by ISS (FC+ISS).

For ML, we estimated 95% confidence intervals (CI) by the Wald method, and for FC, by penalized likelihood profiles [5]. ML and FC estimation was performed using the **logistf** [16] package in R, version 3.5.0. We checked for the presence of separation by the algorithm [17] implemented in the **brglm2** [18] package.

Estimands: The true regression coefficient $\beta_{1}$ was the estimand in our study as $X_{1}$ was considered the target covariate (e.g., exposure to a risk factor). $X_{1}$ was simulated as an unbalanced binary covariate with expected value 0.8.

Performance measures: As for prediction, separation might not necessarily be considered as a problem; we focused on estimation and evaluated bias ($E\left({\hat{\beta }}_{1}-\beta_{1}\right)),\;$ and mean squared error (MSE; $E{\left({\hat{\beta }}_{1}-\beta_{1}\right)}^{2}$) of regression coefficient estimates as well as the probability that a 95% CI excludes $\beta_{1}=0$ (type I error rate or power), the probability that it includes the true value of $\beta_{1}$ (coverage) and the width of the 95% CIs. To compare the performance of ML+ISS and FC, we defined the cost-adjusted relative efficiency (CARE) of ML + ISS relative to FC as:(4)$$\mathrm{CARE}=\frac{\mathrm{MSE}\left({\hat{\beta }}_{ML+ISS}\right)\times {\bar{N}}_{new}}{\mathrm{MSE}\left({\hat{\beta }}_{FC}\right)\times N},$$
where ${\bar{N}}_{new}$ is the increased sample size needed to remove separation averaged over 1000 simulation runs. This measure weighs the increased efficiency against the costs of increasing the sample size, compared to the efficiency of FC at the original sample size. By its definition, CARE >1 suggests that FC is a more efficient estimator. A similar concept was introduced by Armstrong [19] for a different context.

## 3. Results

This section is divided into two parts: First, we report the results from the simulation study described in Section 2.2.; second, we illustrate the problem by two-real life data examples showing results from preliminary and final analyses. The examples are taken from colorectal cancer screening and from ornithology.

### 3.1. Results of Simulation Study

Before evaluating the performance of the methods, we describe the mean new sample size ${\bar{N}}_{new}$ that was required to remove separation and its dependence on the prevalence of separation in the original data sets of size N. These results are shown by means of a nested loop plot [20,21] in Figure 1. Clearly, the mean sample size ${\bar{N}}_{new}$ was positively correlated with the prevalence of separation. Both ${\bar{N}}_{new}$ and the prevalence of separation were generally higher in scenarios with smaller sample sizes, in scenarios where $E\left(Y\right)=0.1$, and with larger effects of $\beta_{1}$. Moreover, they were lower for scenarios with two covariates compared to scenarios with five covariates. Interestingly, many scenarios with ten covariates had the fewest separated data sets. This might seem counterintuitive since, for a fixed non-separated data set, omitting covariates can never induce separation. However, in our simulation study, scenarios with 5 and 10 covariates do not only differ in the number of covariates but also in the model used to generate the binary outcome.

The bias of ${\hat{\beta }}_{1}$ is shown in Figure 2. For ML+ISS, positive bias was generally observed in scenarios where separation occurred in less than a third of simulated data sets. By requiring samples to be nonseparated, the ML+ISS strategy appeared to correct the small-sample bias of ML (away from zero), and in scenarios with high prevalence of separation, its corrective nature even resulted in some negative bias. Applying FC, a small-sample bias-reduction method indeed yielded almost unbiased regression coefficients in many scenarios; however, in extreme cases with small sample sizes and higher values of $\beta_{1}$ negative bias was non-negligible and could be severe, especially if, in addition, the expected outcome prevalence was only 10%. Generally, bias towards zero increased with higher values of $\beta_{1}$. With the exception of some scenarios with strong effects, applying FC to the increased samples (FC+ISS) additionally increased the bias towards zero compared to FC estimation.

Figure 3 presents the MSE of ${\hat{\beta }}_{1}$ and CARE of ML+ISS compared to FC. For most scenarios, there were no considerable differences in terms of MSE of ${\hat{\beta }}_{1}$ between the methods. However, ML+ISS yielded in some cases much larger sample sizes $N_{new}$ with the average sample size ${\bar{N}}_{new}$ up to 11.8 times higher than the original N and a maximum difference between $N_{new}$ and N of 6424 observations. Therefore, FC was a more efficient estimator in all but four scenarios, and in these four scenarios, CARE was very close to one. Interestingly, even with sample sizes of N=500, FC usually performed slightly better than ML+ISS. In extreme situations with the highest values of $\beta_{1}$ and small sample sizes of N=80, the MSE by FC was poor in absolute terms and could only slightly be improved by ML+ISS. The improvement came at the cost of much higher sample sizes, such that FC was substantially more efficient (CARE >> 1). FC often achieved reasonably small MSE even with large proportions of separation, but it failed considerably in small sample situations (N=80) with strongest effects of $\beta_{1}$ and $E\left(Y\right)=0.1$, and in all small sample situations with $E\left(Y\right)=0.1$ and K=10 covariates included in the model. In the latter situations, ML+ISS as well yielded very unstable regression coefficient estimates with an extremely large MSE. As expected, by increasing the sample size, FC+ISS most often reduced the MSE of FC. However, it also occurred that the MSE was slightly increased despite the larger sample size: In some scenarios where bias towards zero was smaller for FC+ISS than FC, the latter yielded lower MSE, and, notably, in a few scenarios, FC outperformed FC+ISS with respect to both bias and the MSE.

Throughout scenarios with $\beta_{1}=0$ and $E\left(Y\right)=0.25$, the type I error rate for FC was closest to the nominal level of 5%, while by ML+ISS and FC+ISS, the null hypothesis was on average rejected with slightly lower probability. The mean type I error rate over all these scenarios was 4.8% for FC, and 4.6% for ML+ISS and FL+ISS. In scenarios with $E\left(Y\right)=0.1,$ all three approaches tended to be conservative; the type I error rate was on average 3.6% for FC and FC+ISS and 3.3% for ML+ISS. FC+ISS had the largest power throughout scenarios with a nonzero effect of $\beta_{1}$. However, in scenarios with larger sample sizes of N=500, all approaches resulted in similar power, ML+ISS performing slightly worse than FC and FC+ISS whenever $E\left(Y\right)=0.1$. The power of FC was distinctively lower than the power of FC+ISS or ML+ISS in small sample situations with strong effects; throughout all scenarios with a sample size of N=80 and $E\left(Y\right)=0.1,$ its power stayed close to the α level of 5% irrespective of $\beta_{1}$. The mean power to detect $\beta_{1}\neq 0$ over all scenarios with $E\left(Y\right)=0.25$ was 49.7% for FC, 49.8% for ML+ISS, and 52% for FC+ISS, and for $E\left(Y\right)=0.1,$ the mean power was 25.9% for FC, 33.7% for ML+ISS, and 38.2% for FC+ISS, respectively. See Appendix A for detailed results on power and type I error rate for $\beta_{1}$ over all simulated scenarios.

The empirical coverage levels and the width of the 95% CIs for $\beta_{1}$ are reported in Appendix A, respectively. In scenarios with $E\left(Y\right)=0.25,$ the average coverage of the profile penalized likelihood CIs for FC and FC+ISS was close to nominal. For both approaches, there was only one out of 36 scenarios where the coverage was outside the ‘plausible’ interval $0.95\pm 2.57\surd \frac{0.95\cdot 0.05}{1000}\;$, into which the empirical coverage rate of a method with perfect coverage of 0.95 falls with a probability of 0.99 when the coverage is estimated from the analysis of 1000 simulated data sets. The ML+ISS Wald CIs were, by contrast, slightly conservative and outside the plausible range in 7 out of 36 scenarios. The mean coverage (and the average width) over all scenarios with $E\left(Y\right)=0.25$ was 95.5% (3.2) for FC, 96% (3) for ML+ISS, and 95.5% (2.8) for FC+ISS. The estimates obtained in scenarios with $E\left(Y\right)=0.1$ were far less precise than for $E\left(Y\right)=0.25$, with a mean coverage (and average width) of 96% (4.9) for FC, 96% (13.6 and 3.6 after excluding scenarios with N=80 and K=10) for ML+ISS, and 95.4% (3.5) for FC+ISS. Notably, the coverage of the CIs of FC and FC+ISS was out of the plausible range in 9 and the one of ML+ISS in 18 out of 36 scenarios. These CIs for all three methods were generally conservative; however, in two scenarios with N=80 and $\beta_{1}=2.77$ the coverage of the profile penalized likelihood CIs for FC+ISS was below 89%. Throughout all scenarios, the width increased with larger effect sizes of $\beta_{1}$. FC’s CIs were on average the widest, followed by ML+ISS; FC+ISS resulted in the narrowest CIs.

### 3.2. Examples

#### 3.2.1. Bowel Preparation Study

Our first example is a study comparing four different bowel purgatives (A, B, C, and D) and their effect on the quality of colonoscopy. The procedure was called a ‘success’ if the cecum was intubated by the endoscopist, or ‘failure’ otherwise (see [22] for more details). The data set used in a preliminary analysis consisted of 4132 patients and suffered from a separation issue as the cecum of patients that used purgative B, which was the smallest of the four groups, was always successfully intubated. The final data set consisted of 5000 patients, and there was now a single patient in group B whose cecum could not be intubated. Table 2 shows the regression coefficients estimated by ML and FC in the preliminary and the final analysis. All the analyses presented here were adjusted for age and sex of patients. In line with our simulation results, the point estimates for purgative B vs. A by FC decreased after increasing the sample size, and the confidence interval of FC in the final data set was narrower than that of ML.

#### 3.2.2. European Passerine Birds Study

The second example is a study investigating the influence of migration (migratory, nonmigratory) and type of diet (granivorous, insectivorous, and omnivorous) on the presence of intestinal parasites in the feces of European passerine birds (see [23] for more details). The data set used in the initial analysis consisted of 366 birds, and no intestinal parasites were present in any of the 17 granivorous birds. After the principal investigator was advised to ‘bring more data’ in order to solve the separation issue, the final data consisted of 385 birds, and intestinal parasites were present in 2 out of 30 granivorous birds. Table 3 shows the regression coefficient estimates, adjusted for migration, obtained by ML and FC estimation in the preliminary and the final analysis. Again, the FC estimates decreased dramatically after increasing the sample size.

## 4. Discussion

In medical research, studies investigating a binary outcome often focus on estimating the effect of an independent variable adjusted for others by logistic regression, and ML is the most commonly used method to estimate such an effect. However, in the analysis of small or sparse data sets, ML coefficient estimates are biased away from zero and very unstable, or ML estimation is even impossible. This occurs when covariates can perfectly separate the observations into the groups defined by the levels of the outcome variable as in our examples. (In fact, this may also happen when there is no separation in the data set. Rareness of events together with numerical instability or inaccuracy may make fitted probabilities already indistinguishable from zero or one by software packages.) Researchers confronted with log-odds ratio estimates diverging to ±∞ may question the plausibility of their analysis results, as it is not assumed that the effect of a variable can be truly ‘infinite’. Rather, a nonexisting estimate is the result of an extreme small sample bias and a consequence of ‘bad luck in sampling’ [6].

Software packages do not always warn a user of the nonconvergence of the iterative ML fitting procedure, and an alternative estimation method to plug-in as a remedy of the problem is usually not suggested. Although seldom reported, we are aware that especially in observational studies, increasing the sample size is a common practice as it offers the most straightforward and intuitive solution, albeit its properties have not been examined thoroughly yet and were perhaps rated overoptimistically. In this paper, we investigated the empirical statistical properties of such a strategy as opposed to FC, which has been described as an ‘ideal solution to the problem of separation’ [5], making it the most widely available penalty in software packages [12]. We compared the performance of both approaches in terms of MSE and bias of regression coefficients and also incorporated the cost of increased sample size when evaluating CARE of ML+ISS relative to FC applied on the original data set. To better understand the impact of ISS, we additionally included in our study a strategy where the first sample size is increased until separation is removed and then FC is applied.

For finite sample sizes, the sampling distribution of ML regression coefficients consists of a subdistribution of finite values and a proportion of diverging estimates that can be explained as a consequence of an extreme small sample bias. The ML+ISS analysis compensates the bias away from zero of ML estimation by the bias towards zero that is induced by sampling until the extreme bias is removed. This can be exemplified by assuming that the original data set and the added observations are analyzed separately. The FC estimate obtained from the model fitted on the added observations has a strong negative bias: In the colonoscopy study, e.g., ${\hat{\beta }}_{B}$ for the added observations was equal to $-0.84\;\left[-2.34,\;1.41\right]$, while the estimate in the original data was $1.95\;\left[-0.01,\;6.8\right]$. In parallel, FC estimation can be represented by ML estimation on augmented data consisting of the original and pseudo-observations [5,11]. The model fitted on the pseudo observations has ${\hat{\beta }}_{pseudo}=0$. As such, the ML+ISS strategy indeed results in acceptable biases but sometimes comes at the cost of highly inflated sample sizes. FC applied to data sets of (much) smaller sample sizes, by contrast, generally yields almost unbiased estimates. While the bias away from zero in ML estimation is a result of poor applicability of large-sample properties in small samples, the ISS strategy actually adds bias towards zero and so the two biases may cancel each other out. If combined with the unbiased FC strategy, however, ISS leads to a bias towards zero. When the true effect size is very large, FC can be biased towards zero [5], and ISS can amplify this effect, as demonstrated in our simulation study.

In terms of cost-corrected efficiency, the ML+ISS strategy is clearly outperformed by FC applied to the original, pre-planned sample size. Therefore, whenever the event rate is low or exposures are rare, a good advice to a researcher planning a study would be to consider FC as the method of analysis. However, in studies where not only effect estimates but also predictions are of importance, e.g., if differences in outcomes attributable to different covariates should be described not only on a relative scale but also on the absolute scale of event probabilities, which is often more relevant, caution is advised—FC leads to predicted probabilities biased towards 0.5. This is a consequence of the unbiasedness of the linear predictor, which naturally cannot translate into unbiasedness of its nonlinear transformation on the probability scale. While pulling predicted probabilities towards 0.5 is not problematic when the outcome levels are approximately balanced, in situations of rare events, this bias becomes apparent. Recently, two methods to overcome this shortcoming—FLIC and FLAC—were proposed by Puhr et al. [11], which both yield average predicted probabilities equal to the observed event rate. Therefore, the bias in predicted probabilities is no longer of concern if one of those methods is applied. Alternatively, one can resort to Bayesian methods and use weakly informative priors centered around 0 (so-called shrinkage priors) to cope with separation, e.g., Cauchy priors [24] or log-*F* priors [12]. Other more heuristic corrections based on a redefinition of the outcome variable have also been proposed [13]. Despite the usefulness of these other procedures in some instances, unlike FC, these methods are mostly justified by their empirical behavior and less by theoretical considerations. Moreover, they are not invariant to linear transformations of the design matrix. For brevity, we have not included their evaluation in the present paper as head-to-head comparisons with FC were already performed [11]. In further simulations not included in this report, we also investigated the performance of FLAC, log-*F*, and the combined strategies FLAC+ISS and log-*F*+ISS. FLAC and log-*F* showed similar patterns of behavior as FC and can yield MSEs even smaller than FC as previously shown [11]. The addition of ISS to FLAC and log-*F* had the same impact as with FC.

In order to produce valid and informative research results, it is desirable, already at the design stage of a study, to not fully rely on approximations based on large-sample properties of ML methods. Moreover, the possibility of separation, or more generally, of effects of sparsity in discrete outcome data should be taken into consideration and a suitable analysis strategy chosen. Although FC can be seen as an advancement of the traditional ML analysis, it cannot solve all problems of data sparsity [25]. Caution is needed when applying FC in extreme situations where the sample sizes are (too) small and the outcomes unbalanced at the same time: It should be taken into account that in such situations, FC performs poorly and lacks power to detect even very strong effects. Finally, researchers should not believe that collecting more data could save a study after analysis has failed but are advised to consult and involve experienced biostatisticians already at the design stage of their studies. For example, a biostatistician may consider the use of simulation to estimate the precision with which an effect can be estimated at different sample sizes and how the result of the study may influence decision making [26]. In such simulations, meaningful assumptions on the distribution of exposure and confounders could be made. Furthermore, the possibility of separation is adequately considered and the superiority of FC vs. ML as the main analysis strategy could be demonstrated.

## 5. Conclusions

By means of a simulation study and the analysis of two real data examples, we compared two strategies for dealing with separation in logistic regression with respect to their empirical performance. We conclude that sampling observations until the problem of separation is removed have adequate performance in terms of precision and inference but are relatively inefficient compared to Firth’s correction applied to a data set of original size.

## Figures and Tables

**Figure 1 ijerph-16-04658-f001:**
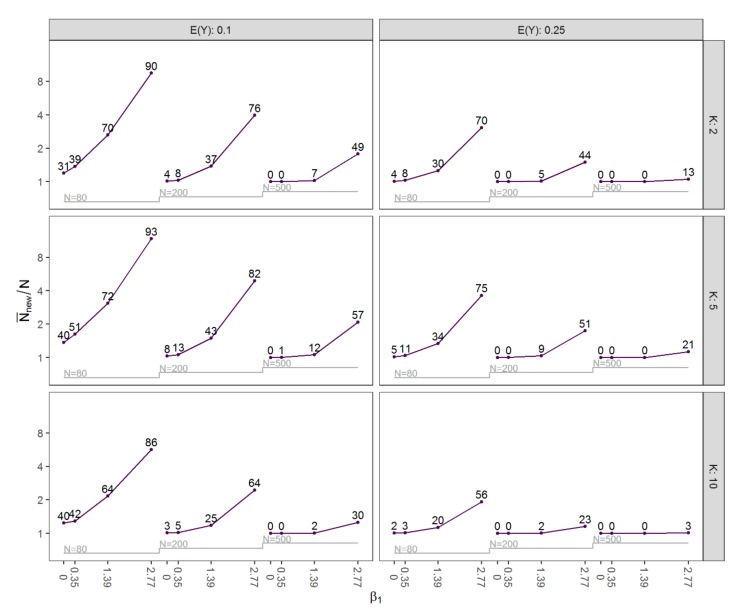
Nested loop plot of ${\bar{N}}_{\mathrm{new}}/N$ by the expected value of Y,
$E\left(Y\right)\in \left\{0.1,\;0.25\right\},$ the number of covariates $K\in \left\{2,\;5,\;10\right\}$, the value of $\beta_{1}\in \left\{0,\;0.35,\;1.39,\;2.77\right\}$, and the sample size $N\in \left\{80,\;200,\;500\right\}$ for all simulated scenarios. The numbers indicate the prevalence of separation (%) with sample size N.

**Figure 2 ijerph-16-04658-f002:**
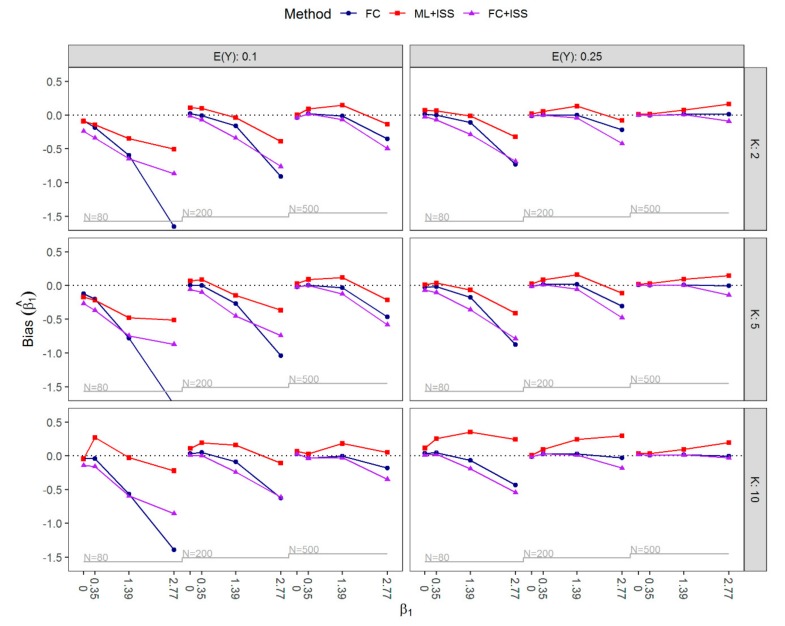
Nested loop plot of bias of ${\hat{\beta }}_{1}$ by the expected value of Y,
$E\left(Y\right)\in \left\{0.1,\;0.25\right\},$ the number of covariates $K\in \left\{2,\;5,\;10\right\}$, the value of $\beta_{1}\in \left\{0,\;0.35,\;1.39,\;2.77\right\},$ and the sample size $N\in \left\{80,\;200,\;500\right\}$ for all simulated scenarios. FC, Firth’s correction; ML+ISS, maximum likelihood combined with the increasing sample size approach; FC+ISS, Firth’s correction combined with the increasing sample size approach.

**Figure 3 ijerph-16-04658-f003:**
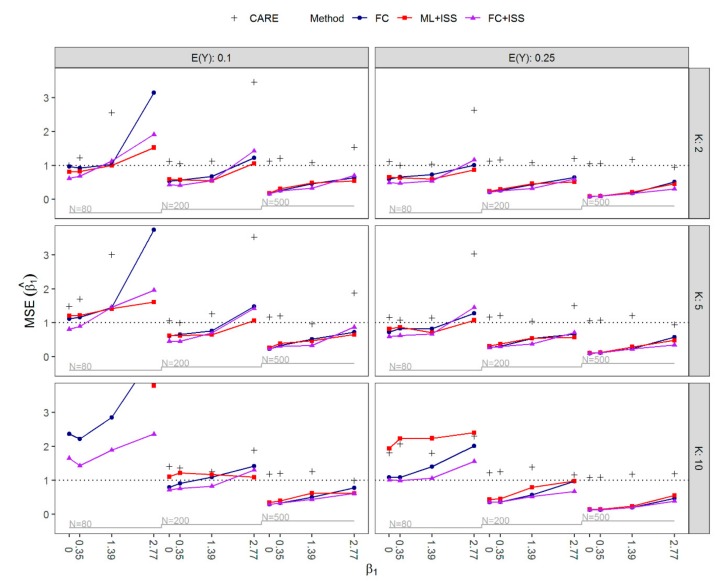
Nested loop plot of the mean squared error (MSE) of ${\hat{\beta }}_{1}$ by the expected value of Y,
$E\left(Y\right)\in \left\{0.1,\;0.25\right\},$ the number of covariates $K\in \left\{2,\;5,\;10\right\}$, the value of $\beta_{1}\in \left\{0,\;0.35,\;1.39,\;2.77\right\},$ and the sample size $N\in \left\{80,\;200,\;500\right\}$ for all simulated scenarios. In addition, CARE as defined in Equation (4) is shown by +, where CARE>1 suggests that ML+ISS is a less efficient estimator than FC. The results for ML+ISS for $E\left(Y\right)=0.1$, K=10 and N=80 are outside of the plot range. The CARE for some scenarios with $E\left(Y\right)=0.1$ and N=80 also lies outside of the plot range. FC, Firth’s correction; ML+ISS, maximum likelihood combined with the increasing sample size approach; FC+ISS, Firth’s correction combined with the increasing sample size approach; CARE, cost-adjusted relative efficiency.

**Table 1 ijerph-16-04658-t001:** Covariate structure applied in the simulation study. $I\left(\cdot \right)$ is the indicator function that equals 1 if the argument is true, and 0 otherwise. $\left[\cdot \right]$ indicates that a non-integer part of the argument is eliminated.

$z_{ik}$	Correlation of $z_{ik}$	Type	$x_{ik}$	E( $x_{ik}$)
$z_{i1}$	$z_{i2}\left(0.6\right),\;z_{i3}\left(0.5\right),\;z_{i7}\left(0.5\right)$	binary	$x_{i1}=I(z_{i1}<0.84)$	0.8
$z_{i2}$	$z_{i1}\left(0.6\right)$	binary	$x_{i2}=I(z_{i2}<-0.35)$	0.36
$z_{i3}$	$z_{i1}\left(0.5\right),\;z_{i4}\left(-0.5\right),\;z_{i5}\left(-0.3\right)$	binary	$x_{i3}=I(z_{i3}<0)$	0.5
$z_{i4}$	$z_{i3}\left(-0.5\right),\;z_{i5}\left(0.5\right),\;z_{i7}\left(0.3\right),$ $z_{i8}\left(0.5\right),\;z_{i9}\left(0.3\right)$	binary	$x_{i4}=I(z_{i4}<0)$	0.5
$z_{i5}$	$z_{i3}\left(-0.3\right),\;z_{i4}\left(0.5\right),\;z_{i8}\left(0.3\right),\;$ $z_{i9}\left(0.3\right)$	ordinal	$x_{i5}=I\left(z_{i5}\geq -1.2\right)+I\left(z_{i5}\geq 0.75\right)$	1.11
$z_{i6}$	$\;z_{i7}\left(-0.3\right),\;z_{i8}\left(0.3\right)$	ordinal	$x_{i6}=I\left(z_{i6}\geq 0.5\right)+I\left(z_{i6}\geq 1.5\right)$	0.37
$z_{i7}$	$z_{i1}\left(0.5\right),\;z_{i4}\left(0.3\right),\;z_{i6}\left(-0.3\right)$	continuous	$x_{i7}=\left[10z_{i7}+55\right]$	54.5
$z_{i8}$	$z_{i4}\left(0.5\right),\;z_{i5}\left(0.3\right),\;z_{i6}\left(0.3\right),\;$ $z_{i9}\left(0.5\right)$	continuous	$x_{i8}=\left[\max0,\;100\exp\left(z_{i8}\right)-20\right]$	138.58
$z_{i9}$	$z_{i4}\left(0.3\right),\;z_{i5}\left(0.3\right),\;z_{i8}\left(0.5\right)$	continuous	$x_{i9}=\left[\max0,\;80\exp\left(z_{i9}\right)-20\right]$	106.97
$z_{i10}$	-	continuous	$x_{i10}=\left[10z_{i10}+55\right]$	54.5

**Table 2 ijerph-16-04658-t002:** Logistic regression coefficient estimates obtained by maximum likelihood (ML) and Firth’s correction (FC) estimation in the preliminary and final analysis of bowel preparation study. All analyses were adjusted for age and sex of the patients.

Data Set	Bowel Purgative	N	${\hat{\beta }}_{ML}$ [95% CI]	${\hat{\beta }}_{FC}$ [95% CI]
Preliminary version $\left(N=4132\right)$	A	2149	reference	
	B	239	not available	1.95 [−0.01, 6.8]
	C	596	0.98 [−0.21, 2.18]	0.85 [−0.13, 2.16]
	D	1148	−0.83 [−1.33, −0.34]	−0.83 [−1.32, −0.34]
Final version $\left(N=5000\right)$	A	2648	reference	
	B	267	1.4 [−0.59, 3.39]	1.01 [−0.62, 2.64]
	C	799	0.83 [−0.11, 1.76]	0.74 [−0.15, 1.64]
	D	1286	−0.83 [−1.28, −0.39]	−0.83 [−1.27, −0.39]

**Table 3 ijerph-16-04658-t003:** Logistic regression coefficient estimates obtained by ML and FC estimation in preliminary and final analyses of the European passerine bird study. All analyses were adjusted for migration.

Data Set	Diet	N	${\hat{\beta }}_{ML}$ [95% CI]	${\hat{\beta }}_{FC}$ [95% CI]
Preliminary version $\left(N=366\right)$	Granivorous	17	reference	
	Insectivorous	274	not available	1.53 [−0.7, 6.43]
	Omnivorous	75	not available	2.17 [−0.02, 7.06]
Final (ISS)version $\left(N=385\right)$	Granivorous	32	reference	
	Insectivorous	276	0.75 [−0.82, 2.33]	0.57 [−0.73, 2.26]
	Omnivorous	77	1.42 [−0.15, 2.98]	1.24 [−0.05, 2.91]

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
