# Peer review of "Bring More Data!—A Good Advice? Removing Separation in Logistic Regression by Increasing Sample Size"

_ijerph, 2019, doi:10.3390/ijerph16234658_

Round 1

Reviewer 1 Report

This is a well-written manuscript that brings statistical rigor to a common problem in statistical practice.  I have several suggestions and comments but did not uncover any fatal flaws.  However, I do not think that all examples are appropriately presented, especially the one on Table 2, which tells me that neither increase in N or use of FC makes a difference: it is a case where clinicians would need to rely on their prior/judgement to make sense of the data for treatment B.  Some of the discussion of bias in FC seems to be off as well, or at least requiring some clarification.  It seems important to further stress that in some cases data separation is the most important result of empirical analysis, especially if it can be interpreted as the result of true state of nature.

The authors ask important question: is more data always needed to solve problems in sample that was obtained?  In the case of separation, the answer surely depends on whether separation is due to sampling variation (i.e. by chance) or is a feature of data-generating process (i.e. real).  Larger sample size can help remove separation if it is by chance or confirm that it is real. (Please soften claim on L51-52 about “separation … may always be removed by increasing sample size”: only if it is an artefact of sampling).   How much certainty can be gained from further sampling depends, in part on how large original sample is.  Intuitively, if the original sample is 10 cases-controls sets, then more data is likely almost always useful.  However, if the original sample is 10,000 case-control sets, then more data will only exaggerate systematic errors given that random errors are already miniscule. So, the question (posed by that authors) seems to become “how much more one can squeeze out of small and troubled (by separation) dataset by analyzing it using a more powerful technique?”.  This is a fair question, especially in those circumstances in public health research where bigger sample size is not an option (a circumstance often overlooked by consulting statisticians).

The concept of CARE is refreshing and well-introduced.  It could have also been applied to bias as was done by (Armstrong, 1996): this is merely a vote “for” the metric and an encouragement to use it more broadly (no need to re-calculate ARE of Armstrong (1996) for this paper, unless the authors see that important improvements can be made to CARE by examining approach of Armstrong (1996)).

I understand that the authors what to shore up importance of logistic regression, but it is admittedly a terrible statistical technique (as per citations by the authors) that only exists because of historical reasons (of beating out Tobit regression more than half a century ago) and inertia in medical curriculum.  Perhaps it should not therefore be called “indispensable” (L33): there are many techniques that are better at predictions.

L67: Is restriction K<N necessary or merely usual?   There care cases in practice where K>N, forcing investigators to select x’s or to reduce dimensionality (e.g. via factor analysis).

L83: Would it not be more precise to say “at least once of the cells has zero observed counts”?

L84: I think that pi_hat in (0,1) is fine but one cannot say that pi in (0,1) is defensible in all cases as per my introduction.  This makes lines 86-87 too strong, only applicable to special, albeit certainly common, case.

L88-93: It is worth mentioning in closing of this paragraph that prior can be picked to be informative, if justified (as is nearly always the case).  This should only help with separation that is believed to be an artefact.

L115: some of the log-OR are rather extreme, such that exp(2.77)=16 is a huge effect, esp. for continuous variable (for reference, ever smoking and lung cancer has OR that is smaller than 16).  It is fine to consider extreme cases, especially to simulate separation, but this seems to be going too far and OR=4 is enough.  This may help simplify presentation and solve the some of the issues with out-of-range values in Fig 2 and 3.

L160-162: Is it really that surprising and interesting that correlation is more important than a number of independent covariates in producing a separation in a sample?

Figure 1: do numbers above points indicate prevalence of separation in sample size of N? If so, please make this explicit in title of the figure.

Figures 2 and 3: I suggest using thicker line to denote N_new>N for the two ISS scenarios.  This would be more intuitive and aid color-blind readers, such as myself.

Figure 3: Please use a different symbol for CARE, such as a larger star (currently it looks like a dot) or open circle to aid readability.  Please also consider adding a dashed line at 1 to denote a boundary above which ML+ISS is less efficient than FC.

I found Figure 4 to be anti-climatic after Figure 3 (it seems to add little).  Although I appreciate its importance for completeness of analysis, perhaps it belongs in supplemental materials, given that they paper is already rather demanding for an average reader of the journal, who is not a statistician.

Example 3.2.1:  This is a case where FC produces uninformative results with N=4132; point estimates are irrelevant and [-0.001, 7] may as well be the same as “not available”.  I think that they authors need to acknowledge this as the case where gains in knowledge could only have been made by increase in sample size, although [-0.6, 3] with FC is hardly impressive and not materially different than [-0.6, 3] with ML for N=5,000.  When I compared FC (N) with ML (N_new), I see [-0.01, 7] vs [-0.6, 3] and think that much has been gained by further sampling, because I am more confident in result that is more precise (absent systematic errors, as assumed): this is in line with classic advice of (Poole, 2001).

Example 3.2.2:  Here it seems that FC helped with N=366 and lack of association was confirmed with N=385 (using MLE).  The CI shrink considerably with ISS, so there is learning with new data that is not obtained with FC alone.  This needs to be discussed in a more nuanced manner, because it again does not support authors main contention.

A suggestion: Why not artificially decrease N for 3.2.1, such that it is materially smaller than N_new, like down to 2500?  This may held show the benefit on FC, more so than smaller ISS.  I am not sure that Example 3.2.2 adds much (more to supplemental materials?).

322-324 and 331: I do not think that FC creates bias towards zero: it uses flat prior and when N is small, the flat prior pulls posterior of log-OR towards the null.  With ISS, the flat prior looses influence.  This is not bias in a traditional sense and the contrast with ML is not appropriate as presented by the authors.   The pull towards log-OR=0 by the prior is always there and will be more noticeable with large beta and small N, but this is what one expects from the procedures: a fair sample from posterior, not a bias.

327-328: One researchers see how precise the estimate is with FC, they than can decide whether that is good enough for them or conduct simulations, as the authors did, to see just what can be gained from “more research”: please see Phillips(Phillips, 2001) for detailed argument.  It is not possible to claim a priori that the research must stop at FC on N.

References

Armstrong BG. (1996) Optimizing power in allocating resources to exposure assessment in an epidemiologic study. Am.J.Epidemiol.; 144: 192-97.

Phillips CV. (2001) The economics of 'more research is needed'. Int.J Epidemiol.; 30: 771-76.

Poole C. (2001) Low P-values or narrow confidence intervals: which are more durable? Epidemiology; 12: 291-94.

Reviewer 2 Report

This is a simulation study reporting results from various scenarios applying the method of maximum likelihood and Firth’s correction on the original data and on enlarged datasets. It is a well written and presented study. I have one comment.

It is unclear to me why the authors did not apply FLIC and FLAC in two example studies, as they state in the paragraph starting in line 325, that in studies in which predictions are of importance, FC leads to biased predicted probabilities.

Round 2

Reviewer 1 Report

I wish to thank the authors for responding to my concerns and suggestions.  More compelling examples would help convinced applied researchers of the argument for FC ahead of further sampling, however, I suspect that consulting statisticians will find the work useful for giving more nuanced advice that may save from needless further sampling in some situations.  An update on the impact of this paper a few years from now may be useful.